# High Light Acclimation Induces Chloroplast Precursor Phosphorylation and Reduces Import Efficiency

**DOI:** 10.3390/plants9010024

**Published:** 2019-12-23

**Authors:** Ahmed Eisa, Katarina Malenica, Serena Schwenkert, Bettina Bölter

**Affiliations:** Department Biologie I, Botanik, Ludwig-Maximilians-Universität, Großhaderner Strasse. 2-4, 82152 Planegg-Martinsried, Germany; Ahmed.Eisa@lmu.de (A.E.); katarina@malenica.net (K.M.); serena.schwenkert@lmu.de (S.S.)

**Keywords:** chloroplast, protein import, protein kinase, protein phosphorylation, Arabidopsis, acclimation

## Abstract

Acclimation is an essential process in plants on many levels, but especially in chloroplasts under changing light conditions. It is partially known how the photosynthetic machinery reacts upon exposure to high light intensities, including rearrangement of numerous protein complexes. Since the majority of proteins residing within chloroplasts needs to be posttranslationally imported into the organelles, we endeavored to study how this important process is regulated upon subjecting plants from pea and *Arabidopsis* to high light. Our results reveal that acclimation takes place on the one hand in the cytosol by differential phosphorylation of preproteins and resulting from the altered expression of the responsible kinases, and on the other hand at the level of the translocation machineries in the outer (TOC) and inner (TIC) envelope membranes. Intriguingly, while phosphorylation is more pronounced under high light, import itself shows a lower efficiency, along with a reduced accumulation of the Toc receptor proteins Toc34 and Toc159.

## 1. Introduction

Plants as sessile organisms strongly rely on their abilities to adapt to fluctuating environmental conditions. The chloroplast as an environmental sensor plays a crucial role in plant acclimation. To maintain and control its protein homeostasis, drastic changes in the chloroplast proteome composition are required. As a consequence of endosymbiosis, the majority of chloroplast proteins are encoded in the nucleus and synthesized in the cytosol as preproteins. The mandatory protein targeting and import mechanisms represent sophisticated checkpoints to control and modulate chloroplast function [1]. After their synthesis, preproteins need to be protected from misfolding or degradation and are translocated in an unfolded state into the chloroplast across the two surrounding membranes through two translocon complexes, i.e., the translocon at the outer membrane of chloroplasts (Toc) and the translocon at the inner membrane of chloroplasts (Tic) [2]. This provides several opportunities for regulation, starting immediately after the translation of preproteins in the cytosol. Preproteins are bound by various chaperones, including HSP70, HSP90, and 14-3-3 proteins [3,4,5], where 14-3-3 binding is preceded by phosphorylation of a subset of preproteins [4]. The formation of these preprotein complexes is thought to ensure the maintenance of a high import competence of the freshly synthesized preproteins and recruitment to the chloroplast membrane. Identification of the responsible dual specificity kinases STY8, STY17, and STY46 provided a tool to investigate the impact of preprotein phosphorylation in vivo, revealing that their impairment results in delayed chloroplast biogenesis [6,7]. Moreover, we recently showed that the kinases are regulated metabolically via an ACT domain [8].

The import process itself is regulated at several levels: Two receptor proteins at the chloroplast surface, Toc159 and Toc34, vary their oligomerization state and thereby affinity to incoming preproteins depending on GTP-binding and hydrolysis [9]. In addition, the oxidation state of cysteines within all known Toc components drastically changes the import efficiency [10]. The same probably holds true for some Tic components such as Tic110 and Tic40 [11]. Moreover, the stromal redox state influences import activity by mediating the dynamic association of redox active components such as Tic62 and Tic32 to the core complex [12,13]. So far, it is not known which environmental signal triggers these regulatory responses.

In this study, we have investigated the reaction of protein targeting and import efficiency in response to high light exposure. Our results indicate that high light influences both cytosolic targeting mechanisms by triggering phosphorylation, as well as the translocation process, which was found to be reduced in response to high light.

## 2. Results

### 2.1. Several Chloroplast Precursors are Phosphorylated by STY8 in Vitro

Several nuclear encoded chloroplast precursors were chosen and overexpressed in *E. coli*. The recombinant proteins were purified via a C-terminal His-tag and subjected to an in vitro kinase assay with purified STY8. Figure 1A shows the autophosphorylated STY8 as well as phosphorylation of all selected precursors. The precursors of ribulose 1,5-bisphosphate-carboxylase small subunit (pSSU), the chlorophyll *a/b* binding protein 1 (pLhcb1), as well as high chlorophyll fluorescence (pHCF101) were previously shown to be phosphorylated within their presequences [7,14]. As a negative control, the mature form of pSSU, which lacks the transit peptide and is consequently not phosphorylated, is shown. For pSSU, the exact phosphorylation sites have been determined [14] and are indicated by asterisks in Figure 1B. In addition, we analyzed the precursors of the oxygen evolving complex 23 (pOE23), a subunit of the NADH dehydrogenase-like complex (pNdhM) as well as the Ferredoxin:NADP(H) oxidoreductase (pFNR). All of these also contain 5–15 serine or threonine residues in their transit peptides (Figure 1B), suggesting that these precursors can also be phosphorylated by STY8, which we could confirm by an in vitro kinase assay (Figure 1A).

### 2.2. High Light Treatment of Arabidopsis Enhances Phosphorylation

In order to investigate the effect of high light intensity on the phosphorylation of precursors, we exposed two-week-old wild-type (WT) *Arabidopsis* plants to 16 h high light. Subsequently, soluble proteins from leaves were extracted and used to conduct an in vitro kinase assay with the selected precursors (Figure 1C). Strikingly, we observed a significant increase in the phosphorylation of all precursors when plants were exposed to high light in comparison to plants that were further kept under normal light conditions. As a negative control, a kinase assay showing the phosphorylation without and with added precursor (pOE23) is shown in Appendix A. The experiments were repeated three times and quantification of the results show that phosphorylation was increased by 125–150%.

### 2.3. STY Kinases are Upregulated on Protein and mRNA Level under High Light

Upon observing increase of precursor phosphorylation levels in high light conditions, we were wondering whether this was due to increased kinase activity and/or whether the expression of the kinases was also affected. To this end, we aimed to analyze the protein levels of STY8 and STY17 by immunoblotting after normal light and high light treatment. Antibodies were raised against both full length kinases STY8 and STY17. Unfortunately, no functional antibody could be obtained for STY46. The obtained antibodies were tested for their specificity using the *sty8 sty46 sty17* mutant, where *STY8* and *STY46* represent null mutations and *STY17* a knockdown generated by RNAi [7]. Indeed, the STY8 antibody recognized a band at the expected size of 62 kDa, which was absent in the mutant. Likewise, the STY17 antibody detected a protein at the expected size, which was reduced in the RNAi knockdown of STY17 to approximately 25%. We therefore concluded that the antibodies are specific for STY8 and STY17, respectively (Figure 2A).

Next, we analyzed the protein levels of STY8 and STY17 after normal light and high light treatment. Interestingly, the protein levels of both kinases increased by 25% after high light treatment. Since we were not able to analyze STY46 on protein level, we performed a quantitative RT-PCR (qRT-PCR) to analyze the transcript levels of all three kinases in response to high light. In line with the elevated protein levels, we observed a significant increase of STY8, STY17, and STY46 transcripts in response to high light. The most drastic upregulation was observed for STY46 with almost five-fold increase in expression followed by STY8 and STY17 (both at approximately three-fold) (Figure 2C).

### 2.4. Import Efficiency into Arabidopsis and Pea Chloroplasts is Reduced in Response to High Light Treatment

We performed imports of selected preproteins into chloroplasts isolated from pea and Arabidopsis, respectively. To this end, we chose pCysP, pFNR, pLHC, and pFBP since their import behavior under different redox conditions has been characterized [13]. Due to the redox state within chloroplasts being one parameter that changes upon high light exposure, we wanted to first analyze preproteins whose behavior under different redox conditions is known. In general, we observed a decreased import efficiency of ^35^S labelled precursors into chloroplasts from high-light-treated pea (Figure 3A) as well as *Arabidopsis* plants (Figure 3B). The degree of import reduction varies between preproteins and plant species from 70–40% (Figure 3A and B, right panels). For some preproteins, i.e., pCysP, the binding was already reduced as demonstrated by the weaker precursor band (p), which is most likely due to the decreased amount of receptor proteins (below and Figure 3C). An exemplary loading control is shown in Appendix A. The respective gels for each import reaction were used for normalization in the quantifications.

### 2.5. High Light Treatment Leads to Reduced Accumulation of Toc159 and Toc34 in Pea

To investigate whether the reduced import rates are influenced by a reduction of components in the import apparatus, we probed isolated chloroplasts from light-treated pea plants with antisera against the Toc complex receptor proteins Toc159 and Toc34 in comparison to chloroplasts isolated from plants out of normal light. Indeed, both proteins were found to be reduced by approximately 50% (Figure 3C). To show equal purity and loading of the isolated chloroplasts, a Coomassie stained gel is shown in Appendix A.

## 3. Discussion

Plants show manifold acclimatory responses to varying light intensities, especially within plastids. A plethora of metabolic pathways as well as photosynthetic activity need to be adapted to changing light conditions, which includes timely provision of enzymes and other proteins. Since most of the chloroplast proteome needs to be posttranslationally imported from the cytosol, this process itself is subject to intense regulation [10,15,16].

In this study, we investigated the effect of high light treatment on two different steps in protein translocation: First, on the phosphorylation of chloroplast preproteins within their transit peptide and second, on the protein transport itself (summarized in Figure 4). Our results clearly show that preprotein phosphorylation increased upon high light treatment, most likely due to upregulation of the responsible STY kinases on RNA and protein level (Figure 1 and Figure 2). It seems that high light induces a signal perceived in the nucleus leading to higher STY kinases RNA expression and concomitantly an increase in kinase proteins. We did not investigate if these kinases also exhibit a higher autophosphorylation leading to a higher activity, which might also contribute to the elevated precursor phosphorylation. The consequence is a higher proportion of phosphorylated preproteins while their total amount in our experiment was kept constant. This observation allows the conclusion that as long as the amount of preproteins is not limiting, kinases phosphorylate more preproteins under high light conditions. The next critical step in protein translocation in vivo is the removal of the phosphate moiety from the transit peptide by an enigmatic phosphatase, since the negative charge interferes with receptor recognition at the chloroplast surface and transport across the envelopes. Failure to do so was shown to almost completely abolish import [14,17].

Intriguingly, the translocation process itself was also affected by high light—import activity was markedly down regulated (Figure 3), regardless of the preprotein studied. All studied precursors were previously demonstrated to react to the stromal redox state represented by the NADPH/NADP^+^ ratio [13]. A more oxidized stroma, i.e., a lower NADPH/NADP^+^ ratio (caused by a relatively higher amount of NADP^+^) caused a tight association of regulatory Tic components with the core translocon and consequently to a higher import efficiency. Raising the amount of absorbable photons by increasing the light intensity leads to higher photosynthetic activity [18] and thereby to a more reduced stroma through pronounced production of NADPH. However, if the light intensity becomes too strong, photoinhibition occurs and the whole process is tuned down [19]. Though we did not determine the stromal redox state in our experimental setup due to it being highly unreliable with isolated chloroplasts, it is tempting to speculate that increased NADPH synthesis under high light at least plays a role in the observed down regulation of import. A second reason is certainly the diminished amount of the receptor proteins Toc159 and Toc34 (Figure 3C).

Moreover, it is feasible to suggest that increased preprotein phosphorylation also reduces import efficiency in vivo. In response to high-light-induced increased phosphorylation, the activity of the required phosphatase—which could not be identified as of yet—may not be sufficient to dephosphorylate the preproteins efficiently, thus slowing down the import process. However, this hypothesis is difficult to analyze in an in vitro import setup using isolated chloroplasts and thus remains to be verified.

The fact that we observed an upregulation of phosphorylation and a downregulation of import rates for all analyzed precursors suggests that import of nuclear encoded preproteins into the chloroplast is generally negatively affected under the investigated conditions. This may contribute to adjusting the photosynthetic performance, for example by downregulation of the PSII antennae size as to avoid over excitation. It was shown that the amount of light harvesting antenna proteins from PSII are indeed down regulated upon high light exposure [20,21,22]. However, whether this holds true for all imported proteins remains to be investigated. A high-throughput analysis would be required to investigate the behaviors of different protein classes. Especially since not all preproteins are phosphorylated, it is still feasible that some proteins that are required under high light conditions (e.g., repair-related proteins such as thioredoxins and LTO1 [23]) are imported at a faster rate. Proteins that are relatively instable such as D1 and members of the Cytb6f complex (petD) and exhibit a high turnover rate also have a higher translation rate under higher light intensities but since these are encoded in the plastome, they do not need to be imported [24].

## 4. Materials and Methods

### 4.1. Plant Material and Light Treatment

*Arabidopsis thaliana* Columbia ecotype (Col-0, WT) and the mutants were grown on soil in a growth chamber under long-day conditions (16 h/8 h day–night cycle, 120 μE m^−2^s^−1^, temperatures of 21 °C/18 °C (light/dark)) for three weeks. Mutant plants, SALK 072890 (*sty8*) and SALK 116340 (*sty46*) lines, and RNAi (*sty17*) were described previously [7].

For import analyses, WT seeds were surface sterilized and grown on ½ Murashige and Skoog medium with 1% (*w/v*) phytoagar and 1% sucrose for three weeks (16 h/8 h day–night cycle, 120 μE m^−2^s^−1^, temperatures of 21 °C/18 °C (light/dark)). Pea plants (P. sativum L., cv. ‘Arvica’, Prague, Czech Republic) were grown on vermiculite for 10 days in a climate chamber s (14 h/10 h day–night cycle, 120 μE m^−2^s^−1^, temperatures of 20 °C/14 °C (light/dark)).

For normal light and high light treatments, plants were either kept in 120 μE m^−2^s^−1^ (normal light) or 500 μE m^−2^s^−1^ (high light) for 16 h, and the temperature was kept constant at 21 °C.

### 4.2. Purification of Recombinant Proteins

The coding regions for STY8, STY46, and STY17 were cloned in the expression vector pET21a^+^ with a C-terminal His-tag (Novagen) and purified as described previously [8]. pOE23, pHCF101, pNdhM, and pFNRL1 were cloned in the expression vector pET21a^+^ with the following primers introducing appropriate restriction sites: pOE23_SacI_fr 5′-AGTCGAGCTCATGGCGTACAGTGCGTGT-3′, pOE23_NotI_rev 5′-CCCACCGCGGCCGCAGCAACACTGAAAGAAGT-3′, pHCF101_SacI_fr 5′-ACTCGAGCTCATGCCGCTTCTTCATCCACA-3′, pHCF101_NotI_rev 5′-CCCACCGCGGCCGCGACTTCGACTGGAGACAA-3′, pNdhM_SacI_fr 5′-AGTGAGCTCATGGTTGCAGCATTC-3′, pNdhM_NotI_rev 5′-TGCTTAGCGGCCGCAGCGTCCTCTTGAGG-3′, pFNRL1_SacI_fr 5′-AGTCGAGCTCATGGCTGCTGCTATA-3′, pFNRL1_SacI_rev 5′-CCCACCGCGGCCGCGTAGACTTCAACATT-3′. Lhcb1 was cloned into pET21d^+^ using the following primers: pLhcb1.3_NheI fr 5′-CGAT GCTAGC GCCGCCT CAACAATG-3′, pLhcb1.3_NotI rev 5′-CGATGCGGCCGCCTTTCCGGGAACAAAGTTG-3′. The constructs for pSSU is described elsewhere [14]. All constructs were expressed in *E. coli* BL21-CodonPlus (DE3)-RIPL cells. For protein purification, cells were resuspended in lysis buffer (50 mM Tris-HCl, 150 mM NaCl, pH 7.4), lysed using a French press, and centrifuged for 30 min at 20,000× *g*. To isolate inclusion bodies, the pellet was washed one time with 20 mL detergent buffer (20 mM Tris pH 7.5, 200 mM NaCl, 1% deoxycholic acid, 1% nonidet P-40, 10 mM ß-mercaptoethanol), twice with Triton buffer (20 mM, Tris pH7.5, 0.5% Triton X-100, 5 mM ß-mercaptoethanol), and twice with Tris buffer (20 mM Tris pH 8.0, 10 mM DTT). Centrifugation was performed at 20,000× g, 4 °C for 10 min. The final pellet was resuspended in 5 mL urea buffer (50 mM Tris pH 8.0, 100 mM NaCl, 7 M urea) and rotated for up to 12 h at RT. After centrifugation at 20,000× g, RT, for 15 min, the supernatant was rotated with 250 μL Ni Sepharose fast flow (GE Healthcare) at RT for 2 h. Subsequently, the recombinant proteins were eluted with increasing concentration of imidazole in urea buffer (50 mM to 500 mM imidazole). Proteins were analyzed by SDS-PAGE.

### 4.3. Phosphorylation Assays

Recombinant kinase or stroma extract was incubated with recombinant substrate in the presence of 3 μCi of [γ-^32^P]-ATP and 2.5 μm ATP in a total volume of 25 μL of kinase buffer (20 mm Tris-HCl, pH 7.5, 5 mM MgCl_2_, and 0.5 mM MnCl_2_). The reaction was carried out for 15 min at 23 °C and stopped by adding 5 μL of SDS sample buffer. The proteins were separated on a 12% SDS-PAGE followed by autoradiography.

### 4.4. SDS-PAGE and Immunoblotting

For immunoblot analysis, either isolated chloroplasts or total soluble proteins were used. To isolate total soluble proteins, *Arabidopsis* leaves were homogenized in homogenization medium (50 mM Tris pH 8.0, 10 mM EDTA, 2 mM EGTA, 10 mM DTT) using liquid nitrogen and an electronic micropestle. The powdered suspension was incubated for 10 min at RT in the dark and centrifuged at 13,000× *g*, 4 °C for 10 min, the supernatant containing the soluble proteins was collected, and protein concentration was determined using Bradford reagent (0.1% Coomassie brilliant blue G-250, 5% ethanol, 10% phosphoric acid). SDS-PAGE and immunoblotting was performed as described [25]. Blots were developed with enhanced chemiluminescence. STY8 and STY17 antisera were raised in rabbits against the full-length proteins (Biogenes, Berlin, Germany). Toc159 and Toc34 antibody generation was described previously [26,27].

### 4.5. Quantitative Real-Time PCR

Total RNA was isolated from several leaves using the Plant RNeasy extraction kit (Qiagen, Hilden, Germany). cDNA was synthesized from 1 μg of RNA (DNase treated) using iScript™ cDNA Synthesis Kit (BioRad, Hercules, CA, USA). All reactions were done in triplicate on three biological replicates. The following oligonucleotides were used: STY8QRT-PCR for 5‘-CCACGGATGGAACTGATGAGT-3‘, STY8QRT-PCR rev 5′-TACACGATCAGGCTTGAGAAA-3′, STY17QRT-PCR for 5′-AAGGTTTAAAAGATGCATTGA-3′, STY17QRT-PCR rev 5′-CATCAGTTCCATCCGTAGGTA-3′, STY46QRT-PCR for 5′- AGGTGCCAGAACGCATGTTCC-3′, STY46QRT-PCR rev 5′-TTGATAGCAACTTCCTGGCTA-3′, RCE1_qpcr_fr 5′-CTGTTCACGGAACCCAATTC-3′, RCE1_qpcr_rev 5′-GGAAAAAGGTCTGACCGACA-3′. The relative abundance of all transcripts amplified was normalized to RCE1 (At4g36800). For quantitative RT-PCR, the SYBR Green real-time PCR master mix (Roche, Basel, Switzerland) was used, and the reaction was performed in a Bio-Rad CFX96 real-time PCR detection system. Forty-five cycles were performed as follows: 1 s at 95 °C, 7 s at 49 °C, 19 s at 72 °C, and 5 s at 79 °C.

### 4.6. Transcription and Translation

All genes of interest were transcribed by means of SP6 polymerase (NEB, England) according to the manufacturer’s instructions. RNA was snap frozen in liquid nitrogen and stored at −80 °C. Translation war performed with the Flexi Retic Kit (Promega, Mannheim, Germany) for 90 min in the presence of ^35^S Met/Cys (Perkin Elmer, Waltham, MA, USA). Translated preproteins were aliquoted, snap frozen, and stored at –80 °C until further use.

### 4.7. Chloroplast Isolation and Protein Import

Pea chloroplasts were isolated as described previously [28]. Chloroplasts equivalent to 10 µg chlorophyll were applied per import, which was conducted in wash buffer containing 3 mM Na-ATP, 10 mM L-Met, 10 mM L-Cys, 20 mM K-gluconate, 10 mM NaHCO_3_, 0.2% (*w/v*) BSA, plus 2 µL translation product for 5 min at 25 °C. The reaction was stopped by addition of one volume of ice-cold washing buffer. Chloroplast were pelleted by centrifugation for 1 min at 1500× *g*, washed once, and finally resuspended in Laemmli buffer.

Arabidopsis plants were harvested from one plate with a razor blade. The leaf material was homogenized with a polytron mixer for 1–2 s in 25 mL of isolation buffer (0.3 M sorbitol, 5 mM MgCl_2_, 5 mM EDTA, 20 mM HEPES-KOH, 10 mM NaHCO_3_ pH 8.0). The homogenate was filtered through one layer of cheese cloth and the procedure repeated three times with the same leaf material. The pooled filtrate was centrifuged for 4 min at 1200× *g*. The supernatant was discarded and pellets were carefully resuspended in residual isolation medium. Resuspended pellets were loaded onto a step Percoll gradient (30%/82% Percoll in 0.3 M sorbitol 5 mM MgCl_2_, 5 mM EDTA, 20 mM HEPES-KOH pH 8.0) and centrifuged at 1500× *g* for 6 min. The lower band containing intact chloroplasts was removed and washed once with at least three volumes of 50 mM HEPES, 0.3 M sorbitol, 3 mM MgSO_4_ pH 8.0 (wash buffer). The final pellets were resuspended in wash buffer depending on their size and chlorophyll concentration was determined according to Arnon [29]. Import was conducted in wash buffer containing 3 mM Na-ATP, 10 mM L-Met, 10 mM L-Cys, 50 mM ascorbic acid, 20 mM K-gluconate, 10 mM NaHCO_3_, 0.2% (*w/v*) BSA, plus 2 µl translation product for 5 min at 25 °C. Afterwards, chloroplasts were pelleted at 1500× *g*, washed once in 200 µL wash buffer, and then resuspended in SDS loading buffer. Proteins were separated on SDS gels, which were Coomassie stained (see Appendix A), then vacuum dried and exposed on a Phosphoimager Screen for 14 h. Screens were analyzed by a Typhoon Phophoimager (GE Healthcare, Uppsala, Sweden) and radioactive bands were quantified with Image Quant (GE Healthcare). For statistical evaluation, the number of chloroplasts from plants treated with normal light was set to 100%, as was the imported protein. All experiments were performed at least three times.

### 4.8. Accession Numbers

STY8 (At2g17700), STY17 (At4g35780), STY46 (At4g38470), pOE23 (At1g06680), pHCF101 (At3g24430), pNdhM (At4g37925) and pFNRL1 (At5g66190), pSSU (AAA34116), pLhcb1 (AT1G29930), CysP (AT5G06290), FNR (P10933), LHC (P27490), FBP (At3g54050)

## Figures and Tables

**Figure 1 plants-09-00024-f001:**
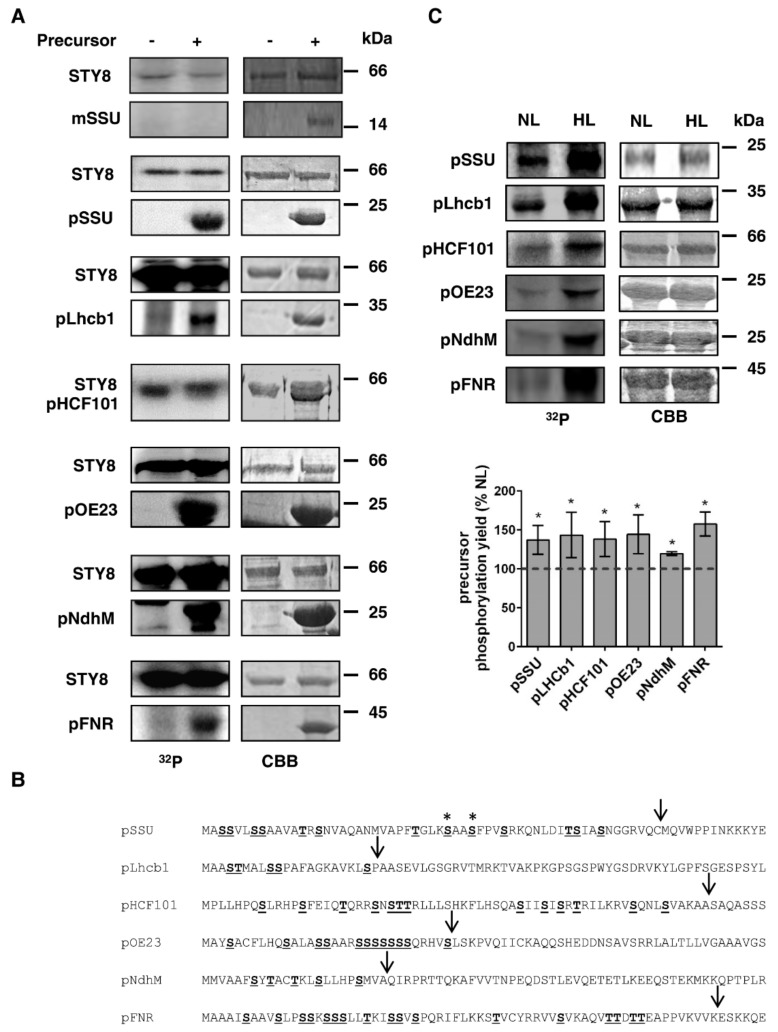
Phosphorylation of chloroplast precursors by STY8 and phosphorylation with plant lysate after high light treatment. (**A**) 1–2 μg of *E. coli* purified recombinant precursors were subjected to an in vitro kinase assay with 0.5 μg *E. coli* purified recombinant STY8 kinase. STY8 autophosphorylation is shown as well as phosphorylation of the substrates. Phosphorylation was detected by autoradiography. A Coomassie stained (CBB) gel is shown as loading control. (**B**) Protein sequences of the precursors used in (**A**) and (**C**). Serine and threonine residues are highlighted. The predicted transit peptide cleavage sites are indicated by arrows. (**C**) Kinase assay showing phosphorylation yield of precursors with plant lysate isolated from wild type (WT) treated with normal light (NL) as control and 16 h high light (HL). Phosphorylation was detected by autoradiography. A Coomassie blue staining (CBB) gel shows equal loading. Quantification of relative precursor phosphorylation intensity after HL treatment is shown. Phosphorylation in NL conditions was set to 100% (indicated by dotted line). Statistical significance was determined by Students t-test. *P* value < 0.05 was considered statistically significant and is indicated by asterisks, *n* = 3.

**Figure 2 plants-09-00024-f002:**
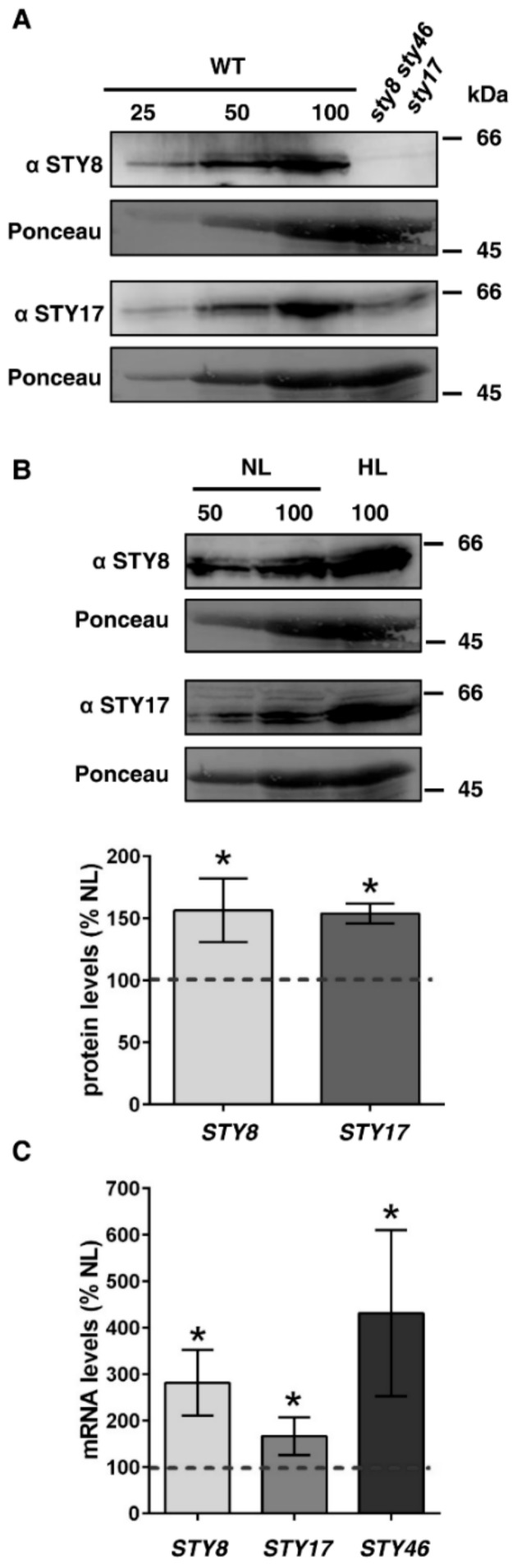
STY kinases are increased on protein and transcript level after high light (HL) treatment. (**A**) WT and *sty8 sty46 sty17* mutants were probed with STY8 and STY17 specific antisera; 80 mg of fresh weight was used to isolate total soluble protein (=100). Ponceau staining shows equal loading. (**B**) WT plants were treated with high light for 16 h and total extracted soluble proteins were probed with STY8 or STY17 antisera. Ponceau staining shows equal loading. The immunoblots were quantified and the relative protein levels are shown. Protein levels after normal light (NL) treatment were set to 100% (indicated by dotted line). Statistical significance was determined by Students t-test. *P* value < 0.05 was considered statistically significant and is indicated by asterisks, *n* = 3. (**C**) qRT-PCR was performed with *STY8*, *STY17*, and *STY46* to analyze the expression level after 16 h HL treatment. Expression levels are shown relative to NL conditions, which were set to 100% for *STY8*, *STY17*, and *STY46*, respectively (indicated by dotted line). Statistical significance was determined by Students t-test. *P* value < 0.05 was considered statistically significant and is indicated by asterisks, *n* = 3.

**Figure 3 plants-09-00024-f003:**
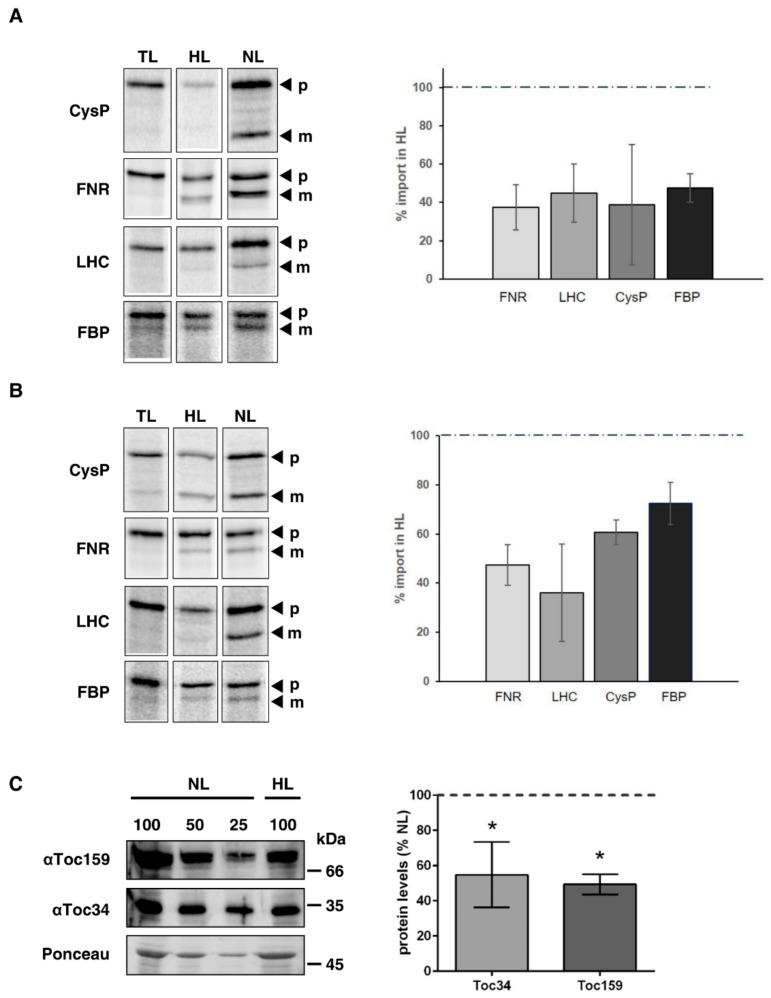
Import into chloroplasts isolated from high-light-treated plants is markedly decreased. (**A**) Chloroplasts were isolated from pea plants treated with 16 h high (HL) or normal light (NL) and used for import experiments with the indicated preproteins (p). Exemplary phosphoimager scans are shown for each preprotein (left). The band representing the respective mature (m) protein after 5 min of import into chloroplasts from normal light treated plants was set to 100% (indicated by the dotted line) (right). (**B**) Chloroplasts were isolated from Arabidopsis plants treated with 16 h high (HL) or normal light (NL) and used for import experiments with the indicated preproteins. Exemplary phosphoimager scans are shown for each preprotein (left). The band representing the respective mature protein after 5 min of import into chloroplasts from normal light-treated plants was set to 100% (indicated by the dotted line) (right). TL represents 10% of the translation product used for the import. (**C**) WT pea plants were treated with high light for 16 h (HL) or kept under normal light conditions (NL). Chloroplast were isolated and probed with Toc159 or Toc34 antisera. Equal amounts of chlorophyll were loaded, shown by Ponceau staining. The immunoblots were quantified and the relative protein levels are shown. Protein levels after normal light (NL) treatment were set to 100% (indicated by dotted line). Statistical significance was determined by Students t-test. *P* value < 0.05 was considered statistically significant and is indicated by asterisks, *n* = 3.

**Figure 4 plants-09-00024-f004:**
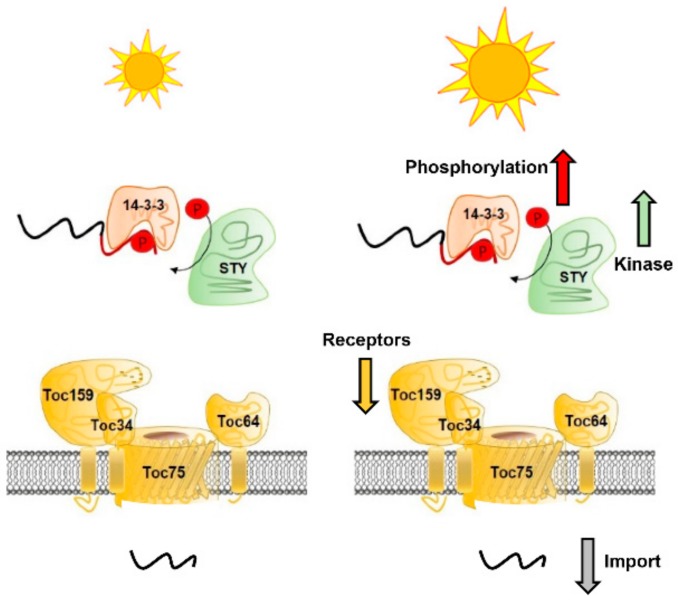
Consequences of high light treatment affecting preprotein import. On the one hand, high light leads to enhanced STY kinase expression as well as preprotein phosphorylation. On the other hand, levels of Toc34 and Toc159 are downregulated and we observed reduced import rates.

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
