# Peer review of "High Light Acclimation Induces Chloroplast Precursor Phosphorylation and Reduces Import Efficiency"

_plants, 2019, doi:10.3390/plants9010024_

Round 1
Reviewer 1 Report
This experiment was conducted on isolated chloroplasts, and it is only explicitly stated in the caption to Fig. 3. This needs to be made clear elsewhere (abstract, results).
The cytosol were synthetic buffers and different in both species, as I could best tell. The “cytosol” was Laemmli buffer for peas. This is not clearly stated and there is no reference for the buffer. The “cytosol” for Arabidopsis was the loading buffer. Again this is not clearly stated.
Author Response
Thank your for taking the time to read our manuscript. Please find our answers to your comments below.
This experiment was conducted on isolated chloroplasts, and it is only explicitly stated in the caption to Fig. 3. This needs to be made clear elsewhere (abstract, results).This is stated e.g. in the results part, page 7, line 160.
The cytosol were synthetic buffers and different in both species, as I could best tell. The “cytosol” was Laemmli buffer for peas. This is not clearly stated and there is no reference for the buffer. The “cytosol” for Arabidopsis was the loading buffer. Again this is not clearly stated.We are not really sure what the reviewer refers to here – kinase or import assays. However, the buffers for each experiment are described in detail in the material and methods section.
Reviewer 2 Report
Eisa et al presents a nice set of data aimed to link the known role of phosphorylation of the transit peptide of nuclear-encoded chloroplast proteins in modulating the import through the TIC-TOC system with the acclimation to increased light conditions. The paper is well written and presented, with a short but comprehensive introduction and compact results and discussion sessions. The results presented are obtained with a variety of techniques which would allow to reach the conclusions drawn in figure 4 if the authors would clarify some of the following points:
1) In figure 1A, it is not clear if OE23, NdhM and FNR should be considered new targets of STY8 or they are already known. If yes, a kinase assay with STY17 or STY46 should be provided to make such statement.
2) Since you see autophosphorylation of STY8 in Figure 1A, in the section 2.2 , figure 1C, are you expecting to see autophosphorylation of the endogenous STY kinases? Could you detect another band with respect to the precursor in the autograph? Especially the band of HCF101, which in figure 1A is very close to STY8, can be considered only HCF101? I suppose a control with P32 and without recombinant precursor added would be beneficial to understand if the kinases are also autophosphorylated.
3) In Figure 3A, CysP precursor signal in HL is lower, why? In general, a loading control in this figure is required. Moreover, what is TL? Is not indicated in the legend nor in the main text.
4) In figure 3C, you show that TOC 159 and 34 are less abundant in HL. Even though a ponceau staining shows equal loading, I think that a control protein that shows no changes should be added. You should indicate if chloroplasts are loaded on chlorophyll basis or protein basis. Moreover, chloroplasts isolated after HL might have a different degree of purity with respect to NL and this might bias the interpretation of the result.
5) Discussion: in lane 192 you don't consider autophosphorylation of STy8 (or other STY) as a potential reason for differential phosphorylation level of the precursors. Moreover, in lane 210 it's not clear to me why you indicate that the translocation process turned out to respond in the opposite way, since in the previous paragraph you say that phosphorylation negatively affects the import.
Minor comments:
lane 77: correct "was well as phosphorylated"
lane 108 correct "gel is shows"
lane 123: mutant, not mutants
in all figures: add MW for all bands, included the one stained with CBB or ponceau to show equal loading. In figure 3, indicate which is the precursor and which is the mature protein. Moreover, although is self-evident that in figure 3 P32 has been added, is not indicated anywhere, not even in M&M.
In section 4.7, substitute "this" with the actual subject of the sentence, otherwise is potentially miss-leading.
Finally, even though the authors have a clear background in the topic presented, I would suggest to add some more references to papers from other groups, since 10 out of 24 references are self-citation.
Author Response
We thank the reviewer for their time and effort to imrove our manuscript. Please refer to our answers below for details.
Eisa et al presents a nice set of data aimed to link the known role of phosphorylation of the transit peptide of nuclear-encoded chloroplast proteins in modulating the import through the TIC-TOC system with the acclimation to increased light conditions. The paper is well written and presented, with a short but comprehensive introduction and compact results and discussion sessions. The results presented are obtained with a variety of techniques which would allow to reach the conclusions drawn in figure 4 if the authors would clarify some of the following points:
1) In figure 1A, it is not clear if OE23, NdhM and FNR should be considered new targets of STY8 or they are already known. If yes, a kinase assay with STY17 or STY46 should be provided to make such statement.
OE23 is a known target of STY8 (Lamberti et al, 2011). NdhM was chosen because it was shown to interact with 14-3-3 proteins in the same manner as pSSU (Fellerer et al, 2011) which is likewise a known target of STY8, and FNR since it engages the same import pathway as pSSU. We agree with the reviewer that performing the kinase assay with all three kinases would provide a more complete picture, but since it was shown in Martin et al (2008) that all three kinases are able to phosphorylate the same subset of preproteins we did not deem this necessary for the present manuscript, which does not focus on the characterization of the individual kinases.
2) Since you see autophosphorylation of STY8 in Figure 1A, in the section 2.2, figure 1C, are you expecting to see autophosphorylation of the endogenous STY kinases? Could you detect another band with respect to the precursor in the autograph? Especially the band of HCF101, which in figure 1A is very close to STY8, can be considered only HCF101? I suppose a control with P32 and without recombinant precursor added would be beneficial to understand if the kinases are also autophosphorylated.
We do not see autophosphorylation of the kinases in plant extract, probably due to their general low abundance. To make that clear, we added Supplemental Fig. 1 where we show that in spite of the presence of some minor phosphorylated bands (=background) the kinase(s) cannot be addressed at the expected size of 62 kDa. In addition, we now include an examplary kinase assay with and without precursor (OE23) in the same supplemental figure.
3) In Figure 3A, CysP precursor signal in HL is lower, why? In general, a loading control in this figure is required. Moreover, what is TL? Is not indicated in the legend nor in the main text.
We added Supplemental Fig. 2A depicting exemplary Coomassie stained gels of imports into Arabidopsis and pea chloroplasts, respectively. Please also note that the quantifications shown in Figure 3 were obtained by normalization on the respective Coomassie gels in all cases. The weaker signal of CysP in HL is most likely due to a lower recognition/binding to the reduced TOC receptors in addition to slower import. This is now stated in the results. TL represents 10% of the translation product which is now stated in the legend.
4) In figure 3C, you show that TOC 159 and 34 are less abundant in HL. Even though a Ponceau staining shows equal loading, I think that a control protein that shows no changes should be added. You should indicate if chloroplasts are loaded on chlorophyll basis or protein basis. Moreover, chloroplasts isolated after HL might have a different degree of purity with respect to NL and this might bias the interpretation of the result.
We agree with the reviewer that a control protein not affected by light treatment would be a good control. However, we find it difficult to predict which protein could serve as such, since any arbitrary chosen chloroplast protein could be affected by the light treatment. Instead, we provided a Coomassie stained gel of the same chloroplasts isolated from normal or high light treated plants used for the immunoblot as an additional loading control (see Supplemental Fig. 2B) Moreover, this demonstrates that the overall protein pattern is identical in both types of chloroplasts, rejecting the hypothesis that HL treatments leads to contaminations in the chloroplast preparations. Also, all chloroplasts were purified over a Percoll step gradient (as described in material and methods), a reliable method to provide highly pure organelles. Gels were loaded on the basis of chlorophyll; this is now stated in the Figure legend.
5) Discussion: in lane 192 you don't consider autophosphorylation of STY8 (or other STY) as a potential reason for differential phosphorylation level of the precursors. Moreover, in lane 210 it's not clear to me why you indicate that the translocation process turned out to respond in the opposite way, since in the previous paragraph you say that phosphorylation negatively affects the import.
Thank you for the suggestions, we expanded and adjusted the text in the discussion accordingly.
Minor comments:
lane 77: correct "was well as phosphorylated" Done
lane 108 correct "gel is shows" Done
lane 123: mutant, not mutants Done
In all figures: add MW for all bands, included the one stained with CBB or ponceau to show equal loading. In figure 3, indicate which is the precursor and which is the mature protein. Moreover, although is self-evident that in figure 3 P32 has been added, is not indicated anywhere, not even in M&M.
We added MW as requested and indicated precursor and mature bands in Fig. 3. For Fig. 3, no 32 P was added – preproteins were labelled with35 S as described in the methods section. For clarification, this is now also mentioned in the results part.
In section 4.7, substitute "this" with the actual subject of the sentence, otherwise is potentially miss-leading. Done
Finally, even though the authors have a clear background in the topic presented, I would suggest to add some more references to papers from other groups, since 10 out of 24 references are self-citation.
We followed the suggestion and added additional references from other research groups so that the manuscript now features 30 citations in total.
Reviewer 3 Report
The presented work is a follow up study of the function of three members of the STY protein kinase family in Arabidopsis. STY proteins were previously shown by the authors to phosphorylate the transit peptides of several chloroplast targeted proteins.
Here, the authors investigate several more precursor proteins which are all targets of STY8. Interestingly, phosphorylation of all precursor proteins investigated significantly increased using extracts of high light treated plants. Thus, the authors show that STY functions are important under high light. Regulation of phosphorylation is likely due to an up-regulation of STY mRNA and protein levels under high light. High light-induced decrease of receptor components of the import apparatus in turn results in a reduced chloroplast import efficiency in pea and Arabidopsis. This all convincingly demonstrates a concerted remodelling of the import processes at different levels under high light important for the regulation of the chloroplast protein import.
The manuscript is scientifically sound and well written. I have only minor issues which need to be addressed.
- line 66: change wording (2 x response): In this study, we have investigated the response of protein targeting and import efficiency in response to high light exposure.
- Fig. 1A: Investigate at least one mature protein or point mutation(s) of the phosphorylation site(s) within the transit peptide as control to confirm that exclusively transit peptides are targets of STY8.
The authors showed that TOC receptor proteins are reduced under high light. What about their RNA levels?
Explain why it makes sense to “generally” reduce import efficiency under high light. What about repair proteins important under high light?
Fig. 3: Indicate precursor and mature protein in panels A and B.
Line 230: remove is: ‘However, this is hypothesis is difficult to analyze…..’
Author Response
we would like to thank the reviewer for taking the time to read and improve our manscript. Please find our comments to your suggestions below.
The presented work is a follow up study of the function of three members of the STY protein kinase family in Arabidopsis. STY proteins were previously shown by the authors to phosphorylate the transit peptides of several chloroplast targeted proteins.
Here, the authors investigate several more precursor proteins which are all targets of STY8. Interestingly, phosphorylation of all precursor proteins investigated significantly increased using extracts of high light treated plants. Thus, the authors show that STY functions are important under high light. Regulation of phosphorylation is likely due to an up-regulation of STY mRNA and protein levels under high light. High light-induced decrease of receptor components of the import apparatus in turn results in a reduced chloroplast import efficiency in pea and Arabidopsis. This all convincingly demonstrates a concerted remodelling of the import processes at different levels under high light important for the regulation of the chloroplast protein import.
The manuscript is scientifically sound and well written. I have only minor issues which need to be addressed.
- line 66: change wording (2 x response): In this study, we have investigated the response of protein targeting and import efficiency in response to high light exposure. Done
- Fig. 1A: Investigate at least one mature protein or point mutation(s) of the phosphorylation site(s) within the transit peptide as control to confirm that exclusively transit peptides are targets of STY8.
The mature protein mSSU was added as a control in Figure 1A as suggested.
The authors showed that TOC receptor proteins are reduced under high light. What about their RNA levels?
We followed the reviewers’ suggestion and analysed the RNA levels (see provided Figure qRT PCR receptors). Interestingly, RNA levels for Toc34 and Toc159 were slightly upregulated under HL (2-fold) in contrast to the proteins. This shows that transcription of the receptors does not seem to be downregulated by retrograde signalling initiated by the high light treatment. Rather, the regulation must occur on a post-transcriptional, translational or posttranslational level, for example by decreased translation or increased degradation. Possibly the slight upregulation is a compensatory reaction. Since this data opens up many new questions, which cannot be answered in the revision of this manuscript but will be addressed in future studies, we would prefer not to include it in the manuscript at this point to prevent confusing the readers. However, if the reviewer or editors are in favour of including the data, we are also happy to do so.
Explain why it makes sense to “generally” reduce import efficiency under high light. What about repair proteins important under high light?
To address this point we have added a new paragraph at the end of the discussion.
Fig. 3: Indicate precursor and mature protein in panels A and B. Done
Line 230: remove is: ‘However, this is hypothesis is difficult to analyze…..’ Done

Round 2
Reviewer 2 Report
Thanks to the authors for the answers provided and the modifications in the text which makes the manuscript much easier to read.